# Women and Stigma: A Protocol for Understanding Intersections of Experience through Body Mapping

**DOI:** 10.3390/ijerph17155432

**Published:** 2020-07-28

**Authors:** Katherine M. Boydell, Jill Bennett, Angela Dew, Julia Lappin, Caroline Lenette, Jane Ussher, Priya Vaughan, Ruth Wells

**Affiliations:** 1Black Dog Institute, University of New South Wales, Sydney 2031, Australia; p.vaughan@unsw.edu.au; 2Art & Design, National Institute for Experimental Arts, University of New South Wales, Sydney 2021, Australia; j.bennett@unsw.edu.au; 3School of Health and Social Development, Faculty of Health, Deakin University, Melbourne 3125, Australia; angela.dew@deakin.edu.au; 4School of Psychiatry and National Drug and Alcohol Research Centre, Faculty of Medicine, University of New South Wales, Sydney 2031, Australia; j.lappin@unsw.edu.au; 5School of Social Sciences, Australian Human Rights Institute, Faculty of Arts and Social Science, University of New South Wales, Sydney 2031, Australia; c.lenette@unsw.edu.au; 6Translational Health Research Institute, Western Sydney University, Sydney 2560, Australia; j.ussher@westernsydney.edu.au; 7School of Psychiatry, Faculty of Medicine, University of New South Wales, Sydney 2031, Australia; ruth.wells@unsw.edu.au

**Keywords:** body mapping, women’s health, intersectionality, embodiment, stigma, arts-based knowledge translation, mental health, disability, refugee, Australia

## Abstract

This paper outlines a research and dissemination protocol to be undertaken with specific groups of marginalised women in Australia. Women impacted by significant mental distress, disability, or refugee status are among society’s most vulnerable and disenfranchised groups. They can experience significant social exclusion, marginalisation and stigma, associated with reduced help seeking, deprivation of dignity and human rights, and threats to health, well-being and quality of life. Previous research has assessed the experiences of discrete groups of women but has to date failed to consider mental health–refugee–disability intersections and overlaps in experience. Using body mapping, this research applies an intersectional approach to identify how women impacted by significant mental distress, disability, and refugee status negotiate stigma and marginalisation. Findings on strategies to cope with, negotiate and resist stigmatised identities will inform health policy and yield targeted interventions informed by much-needed insights on women’s embodied experience of stigma. The women’s body maps will be exhibited publicly as part of an integrated knowledge translation strategy. The aim is to promote and increase sensitivity and empathy among practitioners and policy makers, strengthening the basis for social policy deliberation.

## 1. Introduction

There are 12.6 million women in Australia, accounting for 51% of the nation’s population [1]. Women’s health is a priority for the World Health Organisation (WHO) and the Australian government [2,3]. This includes women with significant mental distress, disability, or refugee status, who are among society’s most vulnerable and disenfranchised groups [4]. These women represent a significant proportion of the Australian population: 22% of people identifying as women experience a serious and persistent mental health disorder each year [5], while 17.3% are recorded as having a disability [6]. Since the year 2000, 220,000 refugees have come to Australia. This includes an increasing number of women considered ‘at risk’, for which there is a separate visa category [7].

This research project responds to this prioritisation of women’s health, and focuses on women in vulnerable circumstances due to the impact of significant mental distress, disability or refugee status. These women can experience high levels of social exclusion, sexual and gender-based violence, marginalisation and stigma. Stigma is a mark separating individuals from one another based on a socially bestowed judgment that some people or groups are tainted [8,9]. It often leads to negative beliefs or stereotypes, the endorsement of those stereotypes as real (prejudice), and a desire to avoid or exclude those who are stigmatised (discrimination) [10]. Marginalisation can be multiple, as disabled and refugee women also have mental health problems. Negative attitudes and experiences of rejection continue to affect the quality of life of marginalised women, who are among the most stigmatised groups in society. Risk factors resulting from stigma and marginalisation are associated with reduced help seeking, deprivation of dignity and human rights, and threats to health, well-being and quality of life [7,11,12,13,14,15,16]. Many women demonstrate resilience and agency, which can lead to achieving positive health outcomes [17]. However, there is a need to enrich our understanding regarding how women negotiate stigma and marginalisation to better inform health and social policy and develop targeted interventions for women. This is the overarching aim of this research project.

Previous research has examined the experiences of specific groups of marginalised women (e.g., refugee women [18,19]) but has failed to consider the multiple identities and areas of disadvantage across mental health–refugee–disability intersections. Little is known about extant gaps in knowledge regarding experiences of marginalisation across groups because of this siloed approach, meaning that the complex needs of many women affected by multiple disadvantages have been continually overlooked. Drawing on an intersectionality framework, this project addresses this gap. This study will use an integrated knowledge translation framework [20], encompassing knowledge generation and dissemination and involving stakeholders throughout the research, to examine points of difference *and* commonalities in experiences of marginalisation and stigma, and explore the intersections of embodied subjectivity of women with significant mental distress, disability and refugee status.

Embodied subjectivity denotes the experience of living in, seeing and experiencing the world from the location of our bodies, always in a social context constructed via interactions [19,21]. There is increased interest in knowledge generated through women’s embodied experience and its relevance to understanding stigma, distress and subjectivity as examined through qualitative methods, primarily interviews [19,22]. Such complex experiences cannot be fully conveyed via textual data alone, and visual methodologies yield critical, engaging, reflexive media for researchers and participants to generate evidence that other methods cannot [23]. This project uses body mapping to generate both visual and textual data capturing women’s embodied experiences of stigma and marginalisation. Body mapping is an arts-based research method. Arts-based research refers to the use of art forms in research [24] to generate, interpret, and/or communicate knowledge. Arts-based research seeks to contribute to the developing interrelated epistemological, theoretical and methodological dialogue regarding art as an approach to inquiry [25]. As described in further detail below, body mapping is ideal for studying the intersections of contextual factors that influence health and well-being, for visually depicting lived experiences of stigma, and as a knowledge translation tool to generate and communicate research findings [26].

## 2. Aims

The overarching objective of this project is to investigate how women impacted by significant mental distress, disability and refugee status negotiate stigma and marginalisation, to inform health policy and yield targeted interventions. We seek to do this by adopting ethical research methods that acknowledge and empower participants as co-producers of knowledge. Aims that support this objective are:1To explore the embodied experience of stigma among women impacted by significant mental distress, disability, and refugee status, to co-create findings on coping strategies, agency, and resistance. This will be achieved by examining the lived experiences of multiple forms of disadvantage for women from these groups.2To empower women impacted by significant mental distress, disability, and refugee status, to be agentic co-creators of knowledge in the research process.3To host a knowledge translation forum with key stakeholders (research team, advisory committee, service providers, policy/decision makers, consumer groups, and women) to synthesise and implement findings and collaboratively develop strategies to address the needs of women impacted by significant mental distress, disability, and refugee status.4To gather public feedback on body mapping exhibitions as evidence of the impact of arts-based knowledge translation strategies to communicate experiences of stigma, and to reduce stigmatising attitudes.

## 3. Materials and Methods

### 3.1. Methodological Framework

This project utilises a feminist intersectionality framework, which recognises that marginalised women may embody multiple and interconnected social categories which both relate to their identity, and connect to characteristics of social structures [27,28,29]. This intersectional framing permits us to acquire a more thorough understanding of stigma experiences [30]. Given our project goals, particularly as they relate to the expression and communication of individual experiences of marginalisation, this intersectional framework will inform our approach to participant engagement and data analysis. Individual participant experiences will be attended to as the project seeks to understand the way multiple social identities, such as gender, cultural identity, dis/ability and sexuality, intersect with macro level structural factors (poverty, racism, sexism, and ableism), resulting in heterogeneous experiences of stigma and marginalisation [31].

### 3.2. Advisory Committee and Community Research Assitants

The project will begin with the recruitment of an advisory committee of community stakeholders, including women with lived experiences from participant communities, service providers, and policy and decision makers. The advisory committee will co-develop, co-analyse and co-evaluate research and knowledge translation activities throughout the project. Peer research assistants representing target community groups will be hired and trained to provide support to the research team and assist in facilitating body mapping sessions. The guidance and support of community members and other stakeholders will ensure that the project is ethical, useful, effective and captures the needs and experiences of the targeted communities.

### 3.3. Sample and Recruitment

Sixty women (aged 18 years and over) will be recruited to take part in the project. Women will identify with, or across, the following community groups in Australia: (1) women experiencing significant mental distress, (2) women with disability, and (3) women with a refugee background. Here, *woman* denotes any person who identifies as being a woman, including transgender, cisgender, non-binary, and gender-diverse individuals. We employ an inclusive definition of *woman* (as a socially produced and performed category) to capture the full spectrum of women’s experiences [32,33,34,35,36,37].

Approximately 20 women will be recruited across each group to attend a series of body mapping workshops. Given the project’s focus on intersectional experiences, it is expected that some participants will self-identify as belonging to more than one group. When this occurs, women will be invited to choose which workshop they would like to attend. If, during the recruitment process, intersectional self-identification is not articulated, purposeful recruitment focusing on women with intersectional experiences will be undertaken. Recruitment will be undertaken in collaboration with local service organisations providing supports to women marginalised by significant mental distress, disability, or refugee status.

The sample size was established using the concept of ‘information power’ as articulated in Malterud et al.’s guide to determining qualitative research sample size. Information power acknowledges that “…the more information the sample holds, relevant for the actual study, the lower” the number of participants required [38]. Here, recruiting participants whose knowledge and lived experiences correlate specifically to the research aims and focus results in an information-rich sample [38]. The recruitment of 60 women across three interrelated body mapping streams will result in a high level of information power. Given our goal to generate detailed, finely grained, qualitative data, the sample size will fulfil our research aims.

## 4. Data Collection

Data will be primarily collected via a series of body mapping workshops. Participants will attend three, two-hour body mapping workshops. Once their body map has been created, the women will participate in an in-depth interview about their body map. As well as providing data for analysis, this interview will be used to create a narrative that will accompany the body map when it is exhibited as part of the knowledge translation strategy. Basic demographic information about participants will be collected during the body mapping sessions (regarding family, age, country of birth, etc.) to assist in describing complexities and intersections of experience.

## 5. Procedure

Body mapping is a process that involves each participant generating a drawn life-sized outline of their body—usually produced by a partner tracing around their body whilst they lie on a large sheet of paper (see Figure 1). Women participating in our research may use a pre-made outline if they prefer. This outline is then filled in by each woman during a creative and reflective process, with the goal of representing felt experience or embodied subjectivity. The process includes a brief meditative exercise to focus on the body and breathing [39]. The production of body maps enables participants to express and symbolise emotions and represent stories about their experiences of the world, their lives, and their bodies via visual representation, producing an image depicting embodied experience at the intersection of multiple forms of marginalisation [40]. Body mapping, like other arts-based research methods, combines the rigor of scientific inquiry with the imaginative and creative aspects of the arts [24], making it an effective method through which to explore complex, sometimes difficult to articulate, ideas and experiences.

Body mapping is uniquely suited to this research [26]. The method has a strong social justice history and is intended to be therapeutic. Past participants have noted beneficial effects associated with body mapping, including feelings of affirmation resulting from the opportunity to reflect on their experiences or personal history, and enjoyment derived from art making in a safe, communal environment [41]. These beneficial effects have the potential to build rapport between participants and researchers, and to create a sense of psychological safety, enabling comfortable and productive data collection to occur. Further, body mapping involves visual stimuli and a collaborative, reflective process, which encourages embodied awareness. Embodied experience is closely intertwined with and reinforces a range of important identities (self, group, gender, culture, health status, etc.). Importantly, the body mapping process minimises power imbalances between researchers and participants [42].

## 6. Data Analysis

Data will be analysed using *thematic discourse analysis* and *visual analysis*. Thematic discourse analysis assumes that our experience and internal constructions of reality are constituted in and through discourse [43]. When analysing body maps and body mapping interviews, the research team will consider common threads and inconsistencies embedded in women’s narratives. A broad thematic level of analysis will be used, as per the six-stage work of Braun and Clarke [44]. Analysis will focus on how each woman constructs, experiences and negotiates embodied stigma, and how they subjectively manage, challenge or resist such stigma.

Once the team identifies initial codes, through a process of collaborative reading and discussion, the data set will be coded using qualitative data software program NVivo 12. Following completion of coding, the data will be grouped together and checked for emerging patterns, for variability and consistency, to identify themes. Thematic interpretation will be achieved via a process of reading and re-reading, reference to relevant literature, and consultation with colleagues. Following identification of themes, data will be interrogated for differences and commonalities within and across categories. In this way, specific discourses (underlying systems of meaning) and their function and effects will be identified in the women’s narratives. This multilayered coding and thematic analysis allow for an integrated and balanced analysis of verbal narratives, field notes and body maps.

Visual analysis will be undertaken following Rose’s critical visual methodology framework [45]. This framework enables analysis of images via three meaning-making sites: (i) production of the image, (ii) the image itself, and (iii) the site(s) where audiences view it. These overlap with three modalities: the technological, compositional, and social. Rose’s series of questions to guide image analysis will be modified for use with drawings to account for the participant, the researcher(s), and the image itself. This method complements our thematic analysis; both approaches are underpinned by self-reflexive, context-driven, thematically focused analyses.

## 7. Integrated Knowledge Translation and Evaluation

Body maps will generate research data *and* be used as tools for knowledge translation (KT). Maps will be made accessible to the public via a series of curated exhibitions (see Figure 2).

Arts-based knowledge translation can expand “…understanding of what counts as evidence, as well as appreciation for the complexity and multidimensionality involved in creating new knowledge” [46]. Knowledge conceived in this expansive way is made more accessible to stakeholders such as policy makers, practitioners, service users and their families [47]. Interdisciplinary collaboration among health, social sciences, humanities, and the arts encourages genuine engagement and fosters agency in decision making and is receiving sustained scholarly interest as an innovative form of KT [17,46,48]. Arts-based inquiry views knowledge as socially constructed, creating open rather than closed texts, and amplifying the voice of those who may be ignored or silenced [24,46]. Exhibiting body maps as part of the project’s KT strategy will create the potential for forms of engagement that promote empathy—in particular enabling policy makers to engage with the subjective viewpoints of individuals who are too often invisible to them. Involving women as co-producers of knowledge using creative arts-based methods has the potential to contribute to their social capital, with women expected to experience feelings of community inclusion, as well as a sense of empowerment and positive self-esteem.

The KT strategy will be evaluated to assess the effectiveness of the body mapping exhibitions as a dissemination tool. Methods used to assess audience engagement and elicit dialogue draw on Lafrenière and Cox’s [49] assessment framework for arts-based research and will include moderated post-exhibit large audience focus groups, researcher observational notes, and audience feedback forms.

Knowledge translation will also take place in the form of a KT forum where research findings and outcomes will be shared. The forum will bring together the advisory committee, service providers, policy and decision makers, stakeholder groups (including NGOs, advocacy organisations and others) and women participants to engage with research findings and consider how they might inform the design of a more responsive system of service and supports. The forum will operate using a deliberative dialogue, a group process that helps to integrate and interpret scientific and contextual evidence to inform policy and practice development. Such dialogues have captured attention because of their potential to address factors influencing how research evidence is used in policy making and in practice [50]. Recruitment for the KT forum will take place through the research team’s established professional networks.

## 8. Discussion

This research project will contribute original data to the evidence base on the lived experiences of women with significant mental distress, disability, and refugee status, addressing the problem of sustained disparities among disadvantaged and vulnerable groups. The opportunity for otherwise socially excluded women to contribute their perspectives on unarticulated complex issues will inform sensitive and responsive policy and service provision.

## 9. Conclusions

This research is well positioned to shift discriminatory health and social care practices to person-centred, relational approaches [51]. It aims to reduce societal misconceptions, stigma and stereotype regarding women with significant mental distress, disability and refugee status. It also promises to significantly contribute to the science of knowledge translation by developing a theoretical base to support the choice, development, and evaluation of arts-based interventions that can improve other areas of social life where stigma is prominent. Communicating and translating embodied knowledge artistically, especially when results are targeted to key audiences and decision makers, offer potential for real change.

## Figures and Tables

**Figure 1 ijerph-17-05432-f001:**
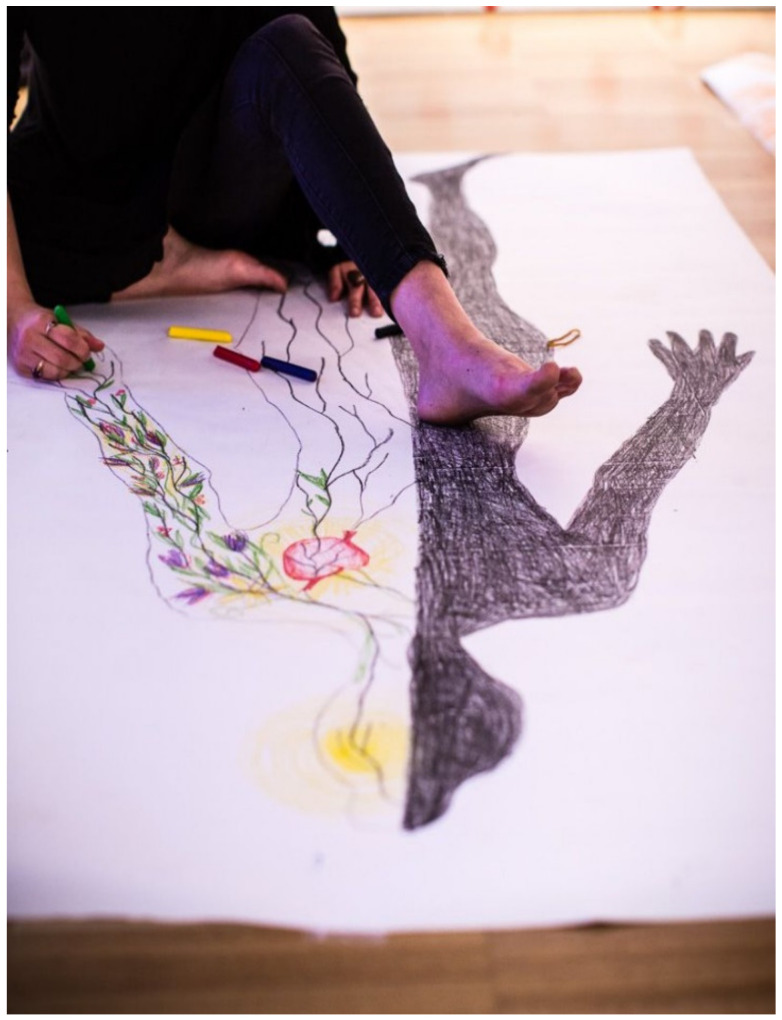
An example of a body map being created. This map was made during a workshop on body mapping anxiety facilitated by team members in Sydney in 2017. Photograph by Diane Macdonald.

**Figure 2 ijerph-17-05432-f002:**
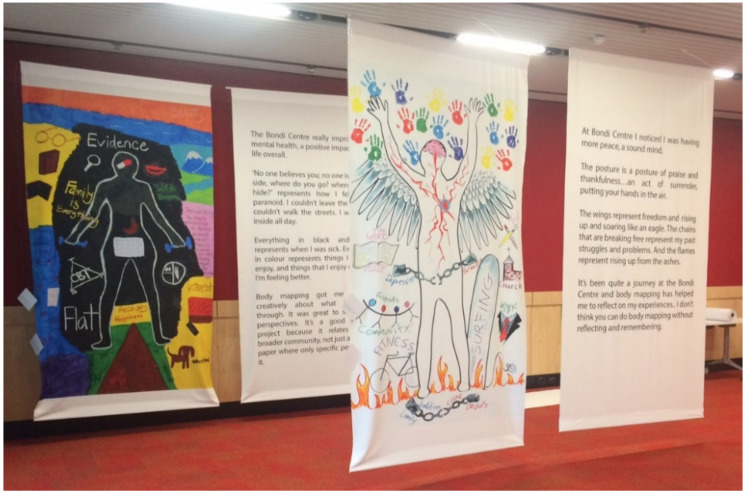
Body maps on display at *Keeping the Body in Mind*, an exhibition co-presented by the Black Dog Mental Health Institute and the National Institute of Dramatic Arts in 2016. Photograph by Katherine M. Boydell.

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
