# Peer review of "Women and Stigma: A Protocol for Understanding Intersections of Experience through Body Mapping"

_ijerph, 2020, doi:10.3390/ijerph17155432_

Round 1

Reviewer 1 Report

I have following comments:

  1. why word women? is it not creating gender inequality?
  2. what age group will beb your study group?
  3. Why do you think that humans who identify themseleves as women (like cisgender) will qualify for your research and not others?
  4. Why you are not focusing on Focus gropu discussions to compliment your analyses?
  5. Family structure, origin of migrant and religion also impacts the behavior hence these need attention?

Reviewer 2 Report

This manuscript presents a novel and interesting content. An Art-based intervention, body mapping, applied to the vulnerable population it has been an interesting topic. However, my role as a reviewer requires a deep analysis of the content and scientific quality of your manuscript. You can find my considerations below:

There are not appropriate headings and subheadings used to help with the organization of the ideas. Manuscripts must contain the required sections: Abstract, Keywords, Introduction, Materials & Methods, Results, Conclusions. Please, check the Journal Instructions for Authors for more details.

One main change is needed to improve the organization of the manuscript. I consider that the information presented in the manuscript is more about the impact of intervention through a qualitative design than the presentation of a therapeutic intervention protocol. I suggest that the authors detail this question precisely and use the appropriate checklist to organized their text:

  • COREQ: Tong A, Sainsbury P, Craig J. Consolidated criteria for reporting qualitative research (COREQ): a 32-item checklist for interviews and focus groups. Int J Qual Health Care. 2007;19(6):349-357 
  • SPIRIT Chan A-W, Tetzlaff JM, Altman DG, Laupacis A, Gøtzsche PC, Krleža-Jerić K, Hróbjartsson A, Mann H, Dickersin K, Berlin J, Doré C, Parulekar W, Summerskill W, Groves T, Schulz K, Sox H, Rockhold FW, Rennie D, Moher D. SPIRIT 2013 Statement: Defining standard protocol items for clinical trials. Ann Intern Med. 2013;158(3):200-207

Page 1, line 2, The title of your manuscript should match the contents of the manuscript and it should identify what the study reports. I suggest “Understanding women's intersections of experience about stigma through Body Mapping" because it could be more concise.

Page 1, line 22, Please, review the abstract section. The abstract should follow the style of structured abstracts, but without headings: 1) Background: Place the question addressed in a broad context and highlight the purpose of the study; 2) Methods: Describe briefly the main methods or treatments applied. 3) Results: Summarize the article's main findings; and 4) Conclusion: Indicate the main conclusions or interpretations. 

Page 1, line 35, Stigma should be included at keywords list

Page 2, line 56, is it the aim of the study?

Page 2, line 64 This sentence should be placed in the methods section

Page 2, line 69 authors said “Given our project goals” but aims are not described yet.

Page 3, line 101 “This will be achieved by examining the lived experiences of multiple forms of disadvantage for women from these groups” should be placed at the methods section

Page 3, line 112 The materials and methods section is written in the future "will" and past tense. Since it is not a project, but a study already carried out, the past should be used.

Page 3, line 113 The first subsection should be the study design. Please place “3.1 Advisory Committee and Community Research Assistants” in the procedure subsection

Page 3, line 142 Please separate data collection and procedure as two distinct subsections.

Page 4, line 182 Give the name and version of any software used and make clear whether the computer code used is available. Include any pre-registration codes.

Página 4, línea 197, La información incluida en "6. Traducción y evaluación integradas de los conocimientos" debe contextualizarse en la sección de introducción. The content provides innovative ideas about therapeutic interventions. However, the review of the literature does not include the use of arts-based research method. Las intervenciones basadas en el arte y la descripción de la literatura deben ser descritas previamente. De esta manera el lector tendrá toda la información necesaria y podrá comprender la intervención en sí misma.

Page 5, Full intervention (who did the intervention, sociodemographic characteristics of participants (age, nationality, condition, …), where was done, individually o by group, how long do the body mapping session and post-interview last, …) details must be provided so that the results can be reproduced If you are showing a new protocol, you should be described in detail while well-established methods can be briefly described and appropriately cited.

Page 5, line 202, Authors could omit “(no p.)”

Page 6, line 234, Findings/ Results section is no included. Findings did not answer the research question, results not described the theme that emerged by thematic discourse analysis and visual analysis.

Page 6, line 235 Authors should discuss the findings and how they can be interpreted in perspective of previous studies and the working hypotheses. The findings and their implications should be discussed in the broadest context possible and limitations of the work highlighted. Future research directions may also be mentioned. This section may be combined with Results.

Reviewer 3 Report

A protocol paper “Women and Stigma: A Protocol for Understanding Intersections of Experience through Body Mapping” proposes  body mapping to document how women under  mental distress, with disability, and having refugee status deal with stigmatization and marginalization. Body mapping is an innovative methodological technique that is often able to capture the imagination of research participants. As a visual technique, it is used to collect qualitative data from participants about their subjective experiences.

The paper is well-written, documented with recent literature and has a potential to enrich our understanding of stigma experiences. Given that it deals with Australia, the title and key words should reflect the geographic.  Also, in introduction, it would be informative to provide at least some data on distribution in Australia of targeted categories of individuals who identify as women: including transgender, cisgender, non-binary, and gender-diverse individuals, especially in refugee group.  

Round 2

Reviewer 2 Report

I am grateful that the authors have analysed, justified, and introduced (or not) the points suggested. In my first review, I did not understand that this study was a protocol. I found confusing information that led me to believe that it had been executed.
Thank you again for applying the changes and discussing in-depth why they have not been applied in the rest.